# Improved Equine Fecal Microbiome Characterization Using Target Enrichment by Hybridization Capture

**DOI:** 10.3390/ani14030445

**Published:** 2024-01-29

**Authors:** Sonsiray Álvarez Narváez, Megan S. Beaudry, Connor G. Norris, Paula B. Bartlett, Travis C. Glenn, Susan Sanchez

**Affiliations:** 1Department of Infectious Disease, College of Veterinary Medicine, The University of Georgia, Athens, GA 30602, USA; connor.norris1@uga.edu (C.G.N.); paulab@uga.edu (P.B.B.); 2Department of Environmental Health Science, College of Public Health, The University of Georgia, Athens, GA 30602, USA; megan.beaudry@uga.edu (M.S.B.); travisg@uga.edu (T.C.G.)

**Keywords:** fecal microbiome, 16S rRNA, TEHC, amplicon sequencing, metagenomics

## Abstract

**Simple Summary:**

Gastrointestinal tract disorders (GITDs) are a serious problem for both adult and young horses. In many cases, the sources of these disorders are unknown, and a link with alterations of the gastrointestinal (GI) microbiome (i.e., GI dysbiosis) is suspected. New alternatives to conventional amplicon sequencing are needed to decipher the real connection between GITDs and dysbiosis. This study compares the performance of target enrichment by hybridization capture (TEHC), a new approach for microbiome sequencing, with conventional amplicon sequencing for the characterization of the equine fecal microbiome.

**Abstract:**

GITDs are among the most common causes of death in adult and young horses in the United States (US). Previous studies have indicated a connection between GITDs and the equine gut microbiome. However, the low taxonomic resolution of the current microbiome sequencing methods has hampered the identification of specific bacterial changes associated with GITDs in horses. Here, we have compared TEHC, a new approach for 16S rRNA gene selection and sequencing, with conventional 16S rRNA gene amplicon sequencing for the characterization of the equine fecal microbiome. Both sequencing approaches were used to determine the fecal microbiome of four adult horses and one commercial mock microbiome. Our results show that TEHC yielded significantly more operational taxonomic units (OTUs) than conventional 16S amplicon sequencing when the same number of reads were used in the analysis. This translated into a deeper and more accurate characterization of the fecal microbiome when the samples were sequenced with TEHC according to the relative abundance analysis. Alpha and beta diversity metrics corroborated these findings and demonstrated that the microbiome of the fecal samples was significantly richer when sequenced with TEHC compared to 16S amplicon sequencing. Altogether, our study suggests that the TEHC strategy provides a more extensive characterization of the fecal microbiome of horses than the current alternative based on the PCR amplification of a portion of the 16S rRNA gene.

## 1. Introduction

Gastrointestinal tract disorders (GITDs) such as colic, colitis, or diarrhea are the leading causes of death in adult horses in the US and rank second as a cause of mortality in foals (young horses) after a combined category of trauma, injury, and wounds [1]. Given the recognized importance of the GIT microbiota in intestinal health in both human and veterinary medicine, several studies have attempted to identify a link between the intestinal microbiota and GITDs in horses [2,3,4,5]. These studies indicated a decrease in GIT bacterial abundance and diversity in horses suffering colic compared to controls but failed to identify specific bacterial species associated with GITD [2,3,4,5,6,7,8,9]. It is suspected that the low taxonomic resolution of the current microbiome sequencing strategies may contribute to the inability to identify the specific bacterial changes potentially associated with colic or other enteric diseases in horses. Hence, new sequencing approaches are needed to obtain an enhanced resolution of the microbiome at the species level.

Since the arrival of next-generation sequencing (NGS), microbiome analyses to underpin certain disease presentations in people and animals are mainly based on the amplification and sequence of a portion of a conserved gene in all bacteria species, the 16S small subunit ribosomal RNA (rRNA) gene [10]. Bacterial 16S rRNA genes contain nine “hypervariable regions” (V1–V9), demonstrating considerable sequence diversity among different bacteria [11]. Most microbiome studies (including equine microbiome studies) are conducted by sequencing and analyzing 16S rRNA regions V1–V3 [12], V3–V4 [4,13,14] or just V4 [15,16,17]. However, no single region can differentiate among all bacteria [11]; therefore, by only looking at a portion of the gene, we can have bacteria species misclassified, identified to the genus level, or even not identified [16,18]. In recent years, metagenomic shotgun sequencing has sought to analyze changes in microbial communities associated with different physiological stages and diseases. This approach aims to sequence the complete genetic content of a sample, including the entire length of the 16S rRNA gene, allowing deeper taxonomic resolution [19]. To our knowledge, metagenomic shotgun sequencing has only been used a couple of times to study the equine microbiome [20,21], most likely due to significant limitations, including the fact that a considerable proportion of the sequenced sample contains non-target DNA (e.g., eukaryotic or microbial DNA not useful for species identification) and costs associated with next-generation sequencing (NGS) [22,23].

Target enrichment by hybridization capture, or TEHC, is a relatively new technique that can help characterize bacterial microbiomes [24,25]. This innovative method combines the most common NGS approaches, 16S rRNA gene amplicons, and shotgun metagenomics (Figure 1). TEHC uses more than 37,000 capture baits targeting the diversity of all bacterial 16S rRNA sequences available in the Greengenes database to enrich metagenomic shotgun libraries. These 120 bp probes are biotinylated. This way, each probe can be recognized by streptavidin-coated magnetic beads for easy extraction and purification of the 16S-mapping DNA sequences [24]. TEHC avoids any bias associated with 16S rRNA PCR amplification and still provides a genomic library carrying mainly 16S rRNA gene sequences. Furthermore, the TEHC genomic library is reported to contain reads covering the entire 16S rRNA gene (not a specific 16S subunit as in the 16S rRNA amplicon approach), translating into a more accurate description of the bacterial communities in a sample [24]. In this study, we compare the performance of TEHC to the conventional 16S amplicon approach to elucidate which technique is more accurate in determining the composition of the bacterial communities in equine fecal samples. This study introduces an improved method for us and others to examine further the interaction between the gut microbiota and equine GI disease.

## 2. Materials and Methods

### 2.1. Sample Collection and DNA Extraction

Four clinical fecal samples from adult horses (>1 year of age) accepted at the Athens Veterinary Diagnostic Laboratory in the University of Georgia (Athens, GA, USA) were used in this study. Microbial DNA was extracted from 1 gr of feces using DNeasy PowerSoil Pro Kit (QIAGEN, Germantown, MD, USA) following the manufacturer’s conditions. DNA stock from the same extraction was used to produce 16S rRNA amplicon and metagenomic libraries for TEHC.

### 2.2. 16S rRNA Amplicon Library Preparation

The hypervariable regions V3 and V4 of the 16S rRNA gene were amplified and sequenced following the 2-step protocol developed by Illumina for preparing 16S rRNA gene amplicons for the Illumina MiSeq system (https://support.illumina.com/documents/documentation/chemistry_documentation/16s/16s-metagenomic-library-prep-guide-15044223-b.pdf (accessed on 23 January 2024)). In the first stage of PCR, V3–V4 regions were amplified with a modified version of the S-D-Bact-0341-b-S-17 and S-D-Bact-0785-a-A-21 24 oligos that also incorporate Illumina adaptors to each amplicon. PCR Illumina indexes were added during the second stage for sample identification after multiplexing. After each PCR step, the resulting amplicons were purified with AMPure XP beads (Beckman Coulter, Indianapolis, IN, USA) following the manufacturer’s protocol at a volume ratio of 1:1. Final library concentrations were measured with Qubit Fluorometric Quantification (ThermoFisher Scientific, Waltham, MA, USA), adjusted to a concentration of 4 nM and pooled.

### 2.3. Target Enrichment by Hybridization Capture (TEHC)

Shotgun metagenomic libraries were prepared using the DNA from the fecal samples and NEBNext^®^ Ultra™ II FS DNA Library Prep Kit for Illumina (New England Biolabs, Ipswich, MA, USA) following the manufacturer’s protocol. After library preparation, the samples were pooled in equimolar ratios and enriched using the myBaits kit (Arbor Biosciences CAT # 308616, Ann Arbor, MI, USA) and the designed 16S rRNA capture baits following the manufacturer’s protocol (v5.01) and as described in the study conducted by Beaudry and collaborators [24]. The final library concentration of the pool was measured with Qubit Fluorometric Quantification (ThermoFisher Scientific, Waltham, MA, USA) and adjusted to a concentration of 4 nM.

### 2.4. Sequencing and Data Analysis

Sequencing was performed in an Illumina MiSeq platform (Athens Veterinary Diagnostic Laboratory, Athens, GA, USA) with 150 bp paired-end reads. Additionally, the reads of the BEI commercial standard genomic DNA mock community HM-276D (BEI Resources, Manassas, VA, USA), sequenced using TEHC (NCBI run SRR13361283) and 16S amplicon (NCBI run SRR13361299) used in a previous study [24], were analyzed and used as a control. HM-276D contains a genomic DNA mixture from 20 bacterial species containing equimolar 16S rRNA gene counts. The 20 species are present in 5% relative abundance and cover 17 genera (https://www.beiresources.org/Catalog/otherProducts/HM-276D.aspx; accessed on 1 March 2022). Raw reads were quality checked using FastQC [26] (v0.11.5) and trimmed and quality filtered using Trimmomatic [27] (v0.36). Filter reads were normalized to have the same number in TEHC and 16S amplicon samples using Seqtk (v1.2 [https://github.com/lh3/seqtk], accessed on 1 March 2022). Read conversion to OTUs was performed as described in the study conducted by Beaudry and collaborators [24]. Briefly, the resulting quality-filtered trimmed reads were mapped to the Greengenes full-length 16S rRNA gene database (v13.8) using BBmap28 (v37.78) with a similarity threshold of 97%. A hit was recorded if both paired-end reads matched the same reference sequence in the database with a similarity of >97% or if only one single paired-end read matched the reference with a similarity of >97%. Rarefaction curves were calculated with the rarecurve function of the R package vegan (v2.6-2) and visualized with ggplot2 (v 3.3.6) also in R. Good’s coverage indices (GCI), or the fraction of the reads that appear in an OTU that has been seen more than once, were calculated in R using the formula: GCI = 100 × (1 − F1/N), where F1 is the number of singleton OTUs, and N is the total number of individuals or the sum of abundances for all OTUs. Each sample’s identified genera and species were manually counted from the OTU table, and percentages were calculated in R. The relative abundance was calculated by normalizing feature counts to the total counts of a sample and visualized using ggplot2 (v 3.3.6) in RStudio (R v4.1.3). Measures of α diversity (Shannon and Simpson diversity indices) and β diversity (Bray–Curtis dissimilarity with non-metric multidimensional scaling [NMDS] ordination) were produced using vegan (v2.6-2) and visualized with ggplot2 (v 3.3.6) R packages. Statistical analyses were performed using linear mixed effects for α diversity metrics (Shannon and Simpson diversity indices) in R package nlme (v3.1-158) and permutational multivariate analysis of variance (adonis) using the R package vegan (v2.6-2) for the β diversity metric. The significance was set at *p* < 0.05.

## 3. Results

### 3.1. TEHC Reveals more OTUs Than Conventional 16S Amplicon Sequencing

The sequencing of the four fecal samples yielded a total of 6,486,776 pair-end reads with an average length of 150 bp. TEHC presented a significantly higher number of raw reads (1,038,135 ± 251,106 [mean ± SEM]) than 16S amplicon (583,559 ± 135,045 [mean ± SEM]). However, that difference was reduced following adaptor removal and quality trimming (TEHC: 780,338 ± 190,597; 16S amplicon: 503,226 ± 116,314 [mean ± SEM]). This was also observed with the BEI commercial mock microbiome samples that were part of a previous run [24] (Appendix A). To have a fair comparison between the two sequencing approaches in determining the fecal microbiome, a subset of the reads from each TEHC sample was randomly selected (see Section 2) to match the number of trimmed reads of the corresponding 16S amplicon sample. The number of reads mapping 16S rRNA gene sequences in the 16S amplicon samples (354,568 ± 80,136 [mean ± SEM]; ~71% of total trimmed reads) was slightly higher than in the TEHC samples (328,199 ± 74,995 [mean ± SEM]; ~65% of total trimmed reads). Still, TEHC samples showed a significantly higher number of OTUs (Appendix A). In addition, a rarefaction analysis showed that the number of OTUs observed for the TEHC samples was consistently higher than the number of OTUs for the 16S amplicon samples regardless of the sampling effort or the number of reads used for taxa assignation (Figure 2). The slopes of the TEHC and the 16S amplicon rarefaction curves flattened with an increase in the number of reads, indicating that most OTUs in each sample were found. This finding is also supported by Good’s coverage indices which were above 0.99 for each sample (Appendix A), showing that >99% of the OTUs were covered with the current number of reads used in the analysis.

When the microbial profiles obtained using the two sequencing strategies were compared, both identified approximately 145 genera per sample. That represented around 61% and 70% of the total genera identified with TEHC and 16S amplicon, respectively. At the species level, an average of 57 species were found in common, representing 50% and 74% of the total species identified with TEHC and 16S amplicon, respectively (Table 1). This indicates that neither technique can identify all bacteria genera and species in the samples. However, a higher number of genera and species were exclusively identified using TEHC (~90 genera and ~57 exclusive species per sample) compared to 16S rRNA amplicon sequencing (~63 genera and ~20 exclusive species per sample) (Table 1).

The relative abundance of each taxon in the sample was explored to determine if the genera and species, exclusively identified using either TEHC or 16S amplicon sequencing, were among the most represented bacterial communities in the sample. Figure 3 shows the relative abundance of phyla that constituted > 1% of the total taxa in each sample. Both NGS strategies identified Firmicutes as the most abundant phylum, followed by Bacteroidetes, Verrucomicrobia, and Proteobacteria. However, TEHC included the phyla Tenericutes and Euryarcheota among the most abundant phyla for some samples, while 16S amplicon sequencing did not (Figure 3). The best-represented genera (>1% of taxa identified in each sample) constitute between ~20% and ~45% of all bacteria genera identified in the fecal samples, with no significant differences (in terms of total abundance) between the samples sequenced using different approaches (Figure 4). However, the different sequencing strategies yielded noticeable differences in the microbial profiles of the most abundant genera, with up to four bacterial genera appearing among the most abundant (above the 1% threshold) with one technique but not the other. In most cases, the relative abundances of the differing genera were not far from the 1% threshold, except for the genera *Lysinibacillus*, whose relative abundance differed nearly 10-fold between the two sequencing strategies (Appendix A).

At the species level, only seven species among the four fecal samples (Fibrobacter succinogenes, Streptococcus luteciae, Clostridium variable, Acinetobacter lwoffi, Lysinibacillus boronitolerans, Bacillus muralis, and Bacteroides uniformis) presented relative abundances of taxa above 1% and never passing 10% of all taxa identified in the sample. As seen at the genus level, the composition of the most abundant bacterial species differs between animals and the sequencing strategy used (Figure 5).

The missing species (that were reported above 1% of all taxa using one technique but not the other) were all identified in the samples with relative abundances below but close to 1%, except for *A. lwoffi* and *L. boronitolerans*, whose relative abundances differed between 10 and 10,000-fold (Appendix A). This shows that although the bacterial species exclusively identified using either TEHC or 16S amplicon are a minority fraction of the sample, the sequencing strategy affects the characterization of the microbial communities at all taxonomic levels.

Seeing the marked discrepancies in the relative bacterial abundances yielded via the different sequencing methods, the composition of a mock commercial microbiome was investigated to determine which sequencing approach provided more accurate results. TEHC and 16S amplicon sequencing identified 15 of the 17 genera comprising the mock sample, missing genera *Escherichia* and *Listeria* (Table 2). Still, as observed with the fecal samples, there were notable differences in the bacterial composition between the different sequencing methods. TEHC recognized 14 out of the 15 genera at the expected relative abundances, while 16S amplicon only matched the expected relative abundances in 9 genera. In most cases, the relative abundances obtained using 16S amplicon sequencing were underestimated, as seen for the genera Lysinibacillus in the fecal samples. Furthermore, 16S amplicon identified an additional genus, *Prevotella*, as present in the control microbiome among the genera with relative abundances above 1%. *Prevotella* was also found when the commercial microbiome was sequenced with TEHC, indicating that this genus could be a potential environmental contaminant whose DNA was amplified during the PCR amplification step of the 16S amplicon library prep, reaching relative abundances above 1% in the sample. Only 5 of the 20 expected species were correctly identified, independent of the sequencing strategy (Appendix A). This indicates that most of the reads were accurately classified at the genus level but not at the species level. Surprisingly, both TEHC and 16S amplicon sequencing identified the Neisseria species at the correct relative abundance but as N. cinerea instead of N. meningitidis. Taken together, these results reinforce the idea that neither technique can provide an accurate taxonomic classification at the species level, but that TEHC gives a more precise characterization of the bacterial communities, especially at the genus level.

### 3.2. TEHC-Sequenced Samples Present a Richer and more Diverse Microbial Community Composition Than Those Sequenced following the 16S Amplicon Approach

The Bray–Curtis dissimilarity metric was used to estimate the disparity in bacterial communities among the TEHC and 16S amplicon samples (Figure 6). This beta diversity metric showed that the microbial composition of the fecal samples differed significantly between the sequencing approaches (*p* = 0.011) but not between animals (*p* = 0.078), supporting what was observed with the relative abundance analyses above. The interaction of these two factors (sequencing approach and animal) was also insignificant (*p* = 0.915). The alpha diversity, or the diversity within each fecal sample sequenced with either TEHC and 16S amplicon, was compared using the Simpson (Figure 7A) and the Shannon–Wiener (Figure 7B) indices. The alpha diversity analysis showed that all TEHC samples present a significantly richer and more diverse bacterial community composition (*p* = 0.0119 and *p* = 0.0002 for Simpson and Shannon–Wiener, respectively), indicating that the TEHC strategy may provide a deeper characterization of the bacterial microbiome of each sample. The alpha diversity analysis also corroborates what was observed above by looking at the number of different genera and species identified in the fecal samples using each technique (Table 1).

## 4. Discussion

There is currently a wide selection of NGS-based strategies for microbiome analysis to choose from. To begin with, NGS technologies can be divided into two main categories: (i) short-read (second-generation) sequencing, and (ii) long-read (or third-generation) sequencing. Both short-read and long-read sequencing strategies have advantages and disadvantages. Reaching species identification with short-read sequencing is indeed challenging because the reads resulting from this technology (<600 bp) do not cover the complete length of the 16S rRNA gene (~1500 bp). This could be easily fixed using long-read sequencing because this technology yields longer reads (10,000–25,000 bp average). However, evidence suggests that the higher error rates of third-generation sequencing technologies may affect the resolution of the analysis considerably and, therefore, do not improve the resolution relative to short-read sequencing as much as was suggested by the increase in read length [28]. The latest long-read sequencing platforms promise a reduced error rate and are becoming an appealing option for microbiome analysis but at a considerably higher cost [29,30]. Another aspect to consider is that long-read sequencing requires highly pure and concentrated DNA samples [30], which may not always be achievable. Because of the abovementioned limitations, we explored different short-read sequencing strategies for studying the equine microbiome. Specifically, this study aimed to compare the performance of the TEHC sequencing strategy (that uses capture probes for enriching 16S rRNA gene sequences from a short-read metagenomic library) to 16S PCR amplicon sequencing for studying the equine fecal microbiome. Before our study, Dr. Beaudry and collaborators found a 435-fold increase in reads mapping to the 16S rRNA gene in the TEHC-enriched metagenomic libraries (~60% of total reads) compared to unenriched metagenomic libraries (~0.1% of total reads) [24]. Furthermore, TEHC was the most cost-effective approach in determining the microbiome in mice fecal samples and mock communities [24]. Still, the consistency and sample processing for 16S rRNA genomic libraries varied considerably between different donor species, and it was still unclear if TEHC’s performance will be superior when applied to other fecal samples with high-fiber content such as the ones obtained from equids. Hence, our study focused on comparing TEHC-enriched metagenomic sequencing and 16S amplicon sequencing to determine which technique is more accurate.

First, our analysis examined the number of raw reads generated via both approaches. Even when the genomic libraries of all samples were adjusted to the same initial DNA concentration to be sequenced (6 pM), TEHC unexpectedly presented a significantly higher number of raw reads than 16S amplicon sequencing. Insert size has been reported to affect sequencing performance as shorter DNA fragments bind more efficiently in the flow cell [31]. The 16S amplicon library consists of DNA fragments of the same size (464 bp), the product of the PCR amplification of V3–V4 16S rRNA gene subunits [32]. On the other hand, TEHC genomic libraries are based on the selection of 16S rRNA gene-matching sequences from a metagenomic library composed of a mix of insert sizes obtained from the unfragmented DNA tagmentation. In this study, tagmentation was set up to obtain the majority of reads with an average length of ~500 bp. Still, remanent fragments ranging from 100 to 1000 bp were also observed when the tagmented DNA was run on a 2% agarose gel. Hence, it is reasonable to assert that reads from the TEHC libraries with sizes < 464 bp may have bonded more efficiently than 16S amplicon reads to the flow cell, explaining the difference in the number of reads between the approaches. Regardless, to have a fair comparison between sequencing approaches, a subset of the quality-filtered reads from each TEHC sample was randomly selected to match the number of filtered reads of its corresponding 16S amplicon sample. Although the number of reads mapping 16S rRNA gene sequences in the 16S amplicon samples was slightly higher than in the TEHC samples, the later presented a significantly higher number of OTUs. This was also observed with the commercial control microbiome sequenced in a different run [24]. Rarefaction curves were performed to see if increasing the sequencing depth (the number of reads) in the 16S amplicon samples could reach the OTU numbers observed in the TEHC samples. Our analysis showed that with the current sequencing effort, most of the OTUs were identified in all samples regardless of the strategy used for sequencing. Therefore, the number of OTUs per sample obtained via TEHC would never be reached using 16S amplicon.

However, does this difference have a biological meaning? Moreover, 16S rRNA-mapping reads are clustered into OTUs based on similarity. For our analysis, reads with 97% of sites agreeing in a pairwise sequence alignment were classified as members of the same OTU, with a representative sequence used to assign taxonomy. In the 16S amplicon sequencing approach, each OTU would ideally represent a single bacteria species (since only a particular section of the 16S rRNA gene would be amplified). TEHC allows the selection and sequencing of DNA fragments covering all regions of the 16S rRNA gene [24], resulting in several OTUs representing a single bacteria species. Thus, the marked differences in the number of OTUs between the TEHC and 16S amplicon samples may not signify that more bacterial species were identified using TEHC but that the 16S rRNA gene sequence of the same species has been covered several times with the TEHC approach. Looking at the total number of genera and species identified in each sample via the different approaches, we observed that TEHC consistently identified significantly more genera and species than conventional 16S amplicon sequencing. This suggests that TEHC could identify bacterial species that conventional 16S amplicon did not. The fact that the reads produced with TEHC sequencing are not restricted to the V3–V4 16S rRNA gene subunits may contribute to identifying genera and species that could not be accurately classified by considering these subunits. This is consistent with previous work that demonstrated that no single region of the 16S rRNA gene could be used to differentiate all bacteria species [11]. Still, around 30% of the bacterial genera and species identified in the samples sequenced with conventional 16S amplicon were not observed when the samples were sequenced using the TEHC approach. This is puzzling and could have several interpretations. The sequencing-approach-specific species may reflect contaminant DNA present in the reagents of the different library preparation protocols [33]. This hypothesis is supported because neither species was found in relatively high abundance in the fecal samples. Unfortunately, one of the limitations of our study is that we did not include negative controls due to cost and lack of space in the sequencing cartridge, and this hypothesis could not be tested. However, if we consider that all species (regardless of the sequencing approach) have been identified correctly and are not contaminants, our results indicate that neither TEHC nor 16S amplicon can accurately identify all the species/genera in the sample. The analysis of the commercial microbiome showed that both techniques failed to identify the genera *Listeria* and *Escherichia* and misclassified 13 of the 18 bacteria species (whose genera were identified), further supporting this hypothesis. Previous literature reported that bacterial species could be misclassified or misidentified by exclusively examining a portion of the 16S rRNA gene [16,18]. Hence, another possible explanation is the potential misclassification/misidentification of the species that exclusively appeared in the samples sequenced with the 16S amplicon approach. Future studies will focus on deciphering which of these scenarios applies to what is observed with the sequencing of the fecal microbiome.

This study also showed that the sequencing approach influenced the taxonomic classification of the most abundant taxa in the fecal samples. Ten predominant phyla were identified in the TEHC samples, while only eight were included when the samples were sequenced using 16S V3 V4 amplicon. *Firmicutes*, *Bacteroidetes*, *Proteobacteria*, and *Verrucomicrobia* were the consistently predominant phyla in our samples, irrespective of the sequencing method, together with phyla *Actinobacteria*, *Fibrobacteres*, *Fusobacteria*, and *Spirochaetes* that were also identified via both sequencing strategies among the most abundant phyla (but not in all samples). This composition is consistent with what was observed in previous studies [6,8,13,14,17,34]. Two phyla, *Tenericutes* and *Euryarchaeota*, were identified with above 1% relative abundance in the TEHC samples but not in the 16S amplicon samples. The *Tenericutes* phylum has been previously reported as a minoritarian phylum of the fecal microbiome of healthy horses [5,15]. The *Euryarchaeota* phylum of the kingdom archaea (not bacteria) also represents a minority, and it has been seen to be significantly more present in “hard keeper” horses that struggle to put on weight than in horses in other Equine Keeper Status Scale (EKSS) groups [35]. The proportion of target sequences within a community affects their detection through PCR-based methods. It also appears that low-abundant taxa are being discriminated against through PCR-based microbial community surveys [36]. These observations further support the idea that TEHC, a technique that avoids PCR biases, may allow the sequencing of more DNA sequences corresponding to less abundant phyla, providing a deeper and more accurate equine microbiome characterization. The differences we observed at the phylum level were magnified when compared with the community composition of the samples at the genus and species levels. Based on the results obtained with the characterization of the mock microbiome, we believe that TEHC provided a more accurate characterization of the fecal microbiota. We postulate that these differences between the different sequencing approaches are mainly caused by PCR-related biases that significantly impact the estimates of microbial diversity [37]. Also, it has been reported that when a bacterial taxon represents less than 1% of the target sequences, the amplification reaction barely generates amplicons corresponding to that species [36]. These facts, together with other PCR-associated biases, such as a potential primer mismatch [38], different genome sizes and the 16S rRNA gene copy number per genome [39], or competition between target sequences in the sample [40], could be behind the different bacterial community profiles obtained via the different sequencing methods.

Alpha and beta diversity are standard metrics used to compare the microbial composition between different samples [41]. We explored if the differences in OTUs had also impacted the description of the fecal microbiome using these indices. We observed that the microbiome of the equine fecal samples was found to be richer when sequenced with TEHC compared to 16S amplicon sequencing, once again supporting the idea that TEHC is a better sequencing strategy to characterize the fecal microbiome in horses and perhaps in other species.

## 5. Conclusions

Microbiome studies are important because they increase our understanding on the relationships between the resident microbial communities and their host. Still, the current technologies available to characterize the microbiome present certain limitations and biases that need to be taken into consideration in order to get improved. Even with a limited number of samples, this study has demonstrated that TEHC circumvents some of the biases associated with current microbiota sequencing approaches, providing a more accurate characterization of the bacterial communities comprising the equine fecal microbiome.

## Figures and Tables

**Figure 1 animals-14-00445-f001:**
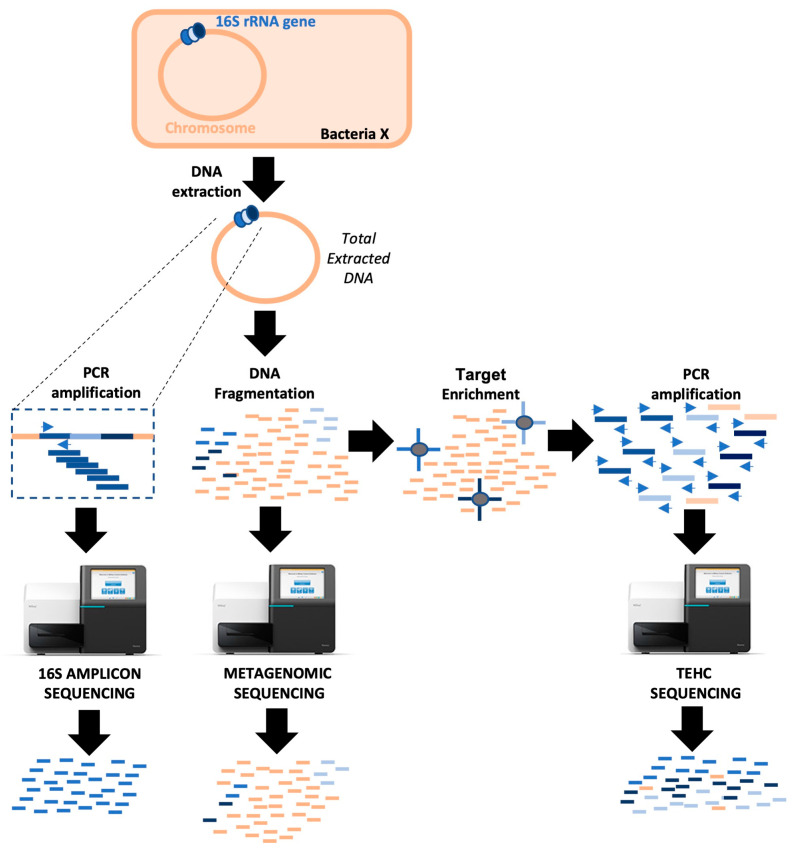
Schematic representation of three different DNA sequencing approaches for microbiome studies. 16S amplicon sequencing is based on the amplification of a portion of the 16S rRNA gene via PCR. This strategy is subject to PCR biases and only provides reads for a specific segment of the 16S rRNA gene, which is not specific enough to differentiate among all bacteria. Metagenomic sequencing is based on the segmentation and subsequent sequencing of total microbial DNA. This technique avoids PCR biases, but less than 3% of the reads are helpful for taxonomic purposes. TEHC combines 16S amplicon and metagenomic sequencing, using specific baits to select 16S rRNA gene sequences from a metagenomic library prior to amplification and sequencing. This strategy reduces PCR biases and provides reads covering all 16S rRNA gene subunits.

**Figure 2 animals-14-00445-f002:**
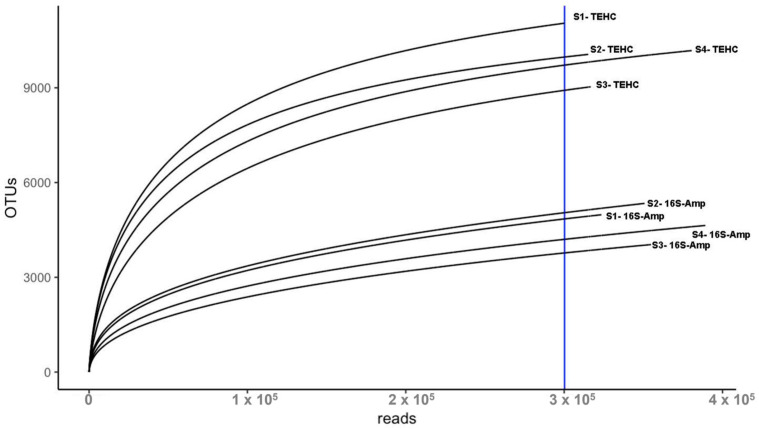
Rarefaction curves illustrate the number of OTUs (Y-axis) identified with increasing numbers of reads (X-axis). S1, S2, S3, and S4 refer to the fecal sample ID; TEHC indicates the samples sequenced with TEHC; and 16S-AMP indicates the samples sequenced using 16S amplicon. The blue vertical line indicates the depth of sampling used to perform the curves.3.2. TEHC Consistently Identified Significantly more Genera and Species Than Conventional 16S Amplicon Sequencing.

**Figure 3 animals-14-00445-f003:**
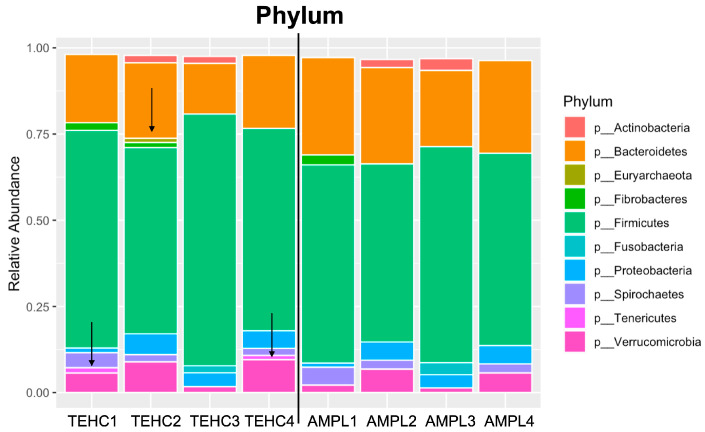
Relative abundance at phylum level. Stacked bar graph of relative abundance of taxa comprising more than 1% of the bacterial sequences in each sample at the phylum level. Colors represent the different phyla observed. Furthermore, 1, 2, 3, and 4 refer to the fecal sample ID; “TEHC” designates the samples sequenced using TEHC; and “AMPL” denotes the samples sequenced using 16S amplicon. Black arrows point at phyla (among the most represented) exclusively found with either of the sequencing approaches.

**Figure 4 animals-14-00445-f004:**
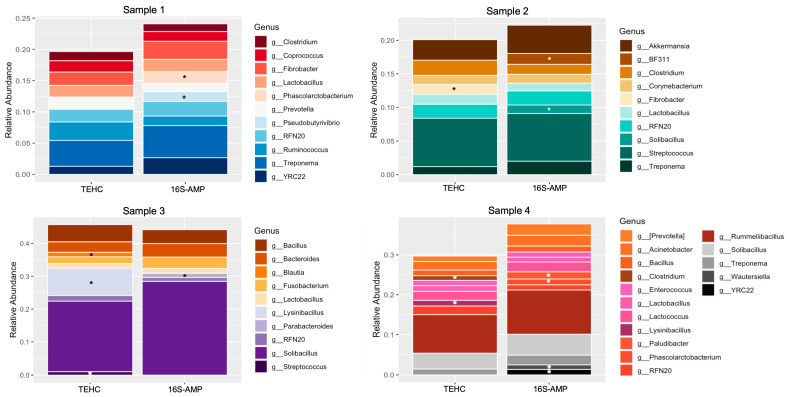
Relative abundance at genera level. Stacked bar graph of relative abundance of taxa comprising more than 1% of the bacterial sequences in each sample at the genera level. Colors represent the different genera observed. “TEHC” denotes the samples sequenced using TEHC, and “16S-AMP” designates the samples sequenced using 16S amplicon. Black and white asterisks point at genera (among the most represented) exclusively found with either of the sequencing approaches.

**Figure 5 animals-14-00445-f005:**
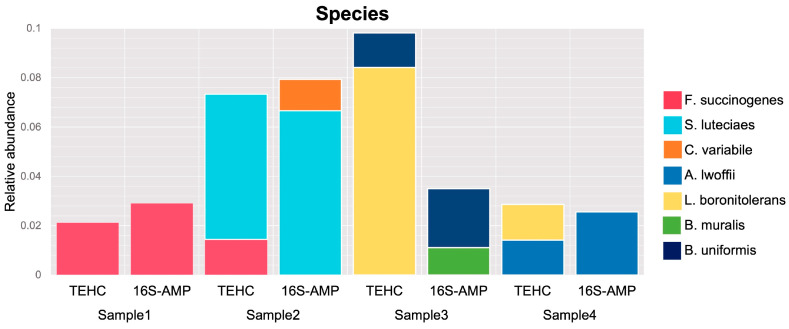
Relative abundance at species level. Stacked bar graph of relative abundance of taxa comprising more than 1% of the bacterial sequences in each sample at the species level. Colors represent the different genera observed. “TEHC” indicates the samples sequenced with TEHC, and “16S-AMP” indicates the samples sequenced using 16S amplicon.

**Figure 6 animals-14-00445-f006:**
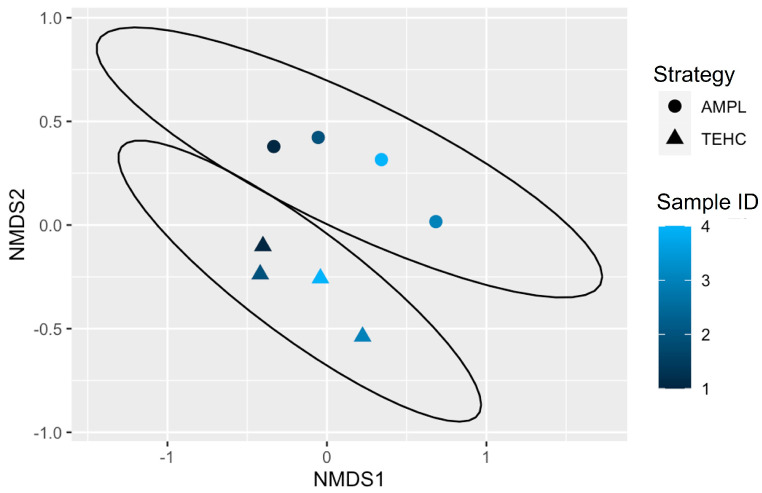
Bray-Curtis dissimilarity with non-metric multidimensional scaling [NMDS] ordination. NMDS1 corresponds to the ordination axis 1 (Y-axis), while NMDS2 corresponds to the ordination axis 2 (X-axis). Shapes indicate different sequencing strategies (circles for 16S amplicon and triangles for TEHC), and colors address the ID of the fecal samples. Calculated ellipses evidence the samples group by sequencing strategy.

**Figure 7 animals-14-00445-f007:**
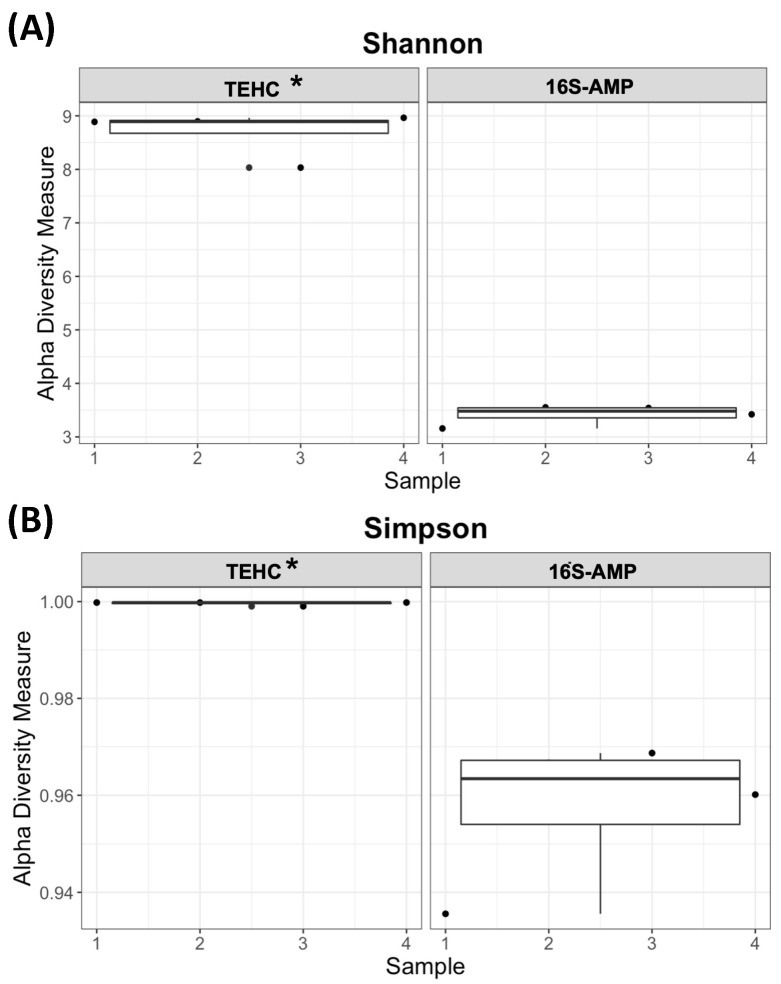
Alpha diversity analysis. Shannon (**A**) and Simpson (**B**) indices of alpha diversity. Furthermore, 1, 2, 3, and 4 refer to the fecal sample ID; TEHC indicates the samples sequenced with TEHC; and 16S-AMP indicates the samples sequenced using 16S amplicon. Asterisks denote significant differences (significance was set at *p* < 0.05).

**Table 1 animals-14-00445-t001:** The number of different genera (A) and species (B) identified with TEHC and 16S amplicon.

(A)						
	Genus level					
	TEHC			16S Amplicon		
Sample	Shared	Exclusive	Total	Shared	Exclusive	Total
1	135 (58%)	99 (42%)	234	135 (72%)	53 (28%)	188
2	143 (60%)	94 (40%)	237	143 (70%)	60 (30%)	203
3	152 (67%)	76 (33%)	228	152 (72%)	58 (38%)	210
4	150 (60%)	99 (40%)	249	150 (67%)	75 (33%)	225
**(B)**						
	Species level					
	TEHC			16S Amplicon		
Sample	Shared	Exclusive	Total	Shared	Exclusive	Total
1	135 (72%)	53 (28%)	188	42 (71%)	17 (29%)	59
2	143 (70%)	60 (30%)	203	54 (73%)	20 (27%)	74
3	152 (72%)	58 (38%)	210	77 (81%)	18 (19%)	95
4	150 (67%)	75 (33%)	225	56 (70%)	24 (30%)	80

“Shared” indicates the number of genera/species identified with both strategies. “Exclusive” refers to the number of genera and species identified with a particular approach exclusively. “Total” is the number of total genera and species identified in a sample. In brackets is the percentage of each number of genera and species present in each sample.

**Table 2 animals-14-00445-t002:** Relative abundance (%) of the genera identified using 16S rRNA amplicon sequencing and TEHC in BEI commercial standard genomic DNA mock community HM-276D.

Genus	16S Amplicon	TEHC	Expected
*Acinetobacter*	6.15	5.50	5
*Actinomyces*	**0.63**	5.08	5
*Bacillus*	**0.04**	**0.18**	5
*Bacteroides*	6.74	6.47	5
*Clostridium*	9.54	7.62	5
*Deinococcus*	3.27	3.72	5
*Enterococcus*	3.94	4.88	5
*Escherichia*	**0.00**	**0.00**	5
*Helicobacter*	**12.19**	5.98	5
*Lactobacillus*	**1.81**	5.25	5
*Listeria*	**0.00**	**0.00**	5
*Neisseria*	7.18	5.11	5
*Propionibacterium*	**0.36**	3.72	5
*Pseudomonas*	**0.22**	4.74	5
*Rhodobacter*	2.49	3.04	5
*Staphylococcus*	10.58	8.71	10
*Streptococcus*	21.77	14.89	15
*Prevotella*	**1.01**	0.01	0

Expected refers to the real composition of the mock microbiome according to the manufacturer’s specification. In bold, relative abundances significantly higher or lower than the expected abundance are shown.

## Data Availability

Raw reads were submitted to NCBI under a bioproject with the accession number PRJNA933177.

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
