# Peer review of "Improved Equine Fecal Microbiome Characterization Using Target Enrichment by Hybridization Capture"

_animals, 2024, doi:10.3390/ani14030445_

Round 1

Reviewer 1 Report

Comments and Suggestions for Authors

This manuscript analyzed the performance of the target enrichment by hybridization capture (TEHC) and 16s rRNA amplicon sequencing in equine fecal microbiome characterization. The authors utilized this method to analyze four equine fecal samples, demonstrating that TEHC reveals more OTUs and richer and more diverse microbial community composition than the 16S-amplicon approach. The authors also used one mock commercial microbiome to unveil that even with relative better accuracy, TEHC fails to detect Escherichia and Listeria in the mock sample. Several critical concerns arise upon assessing this manuscript:

1. The TEHC method was established in a previously published paper “Improved Microbial Community Characterization of 16S rRNA via Metagenome Hybridization Capture Enrichment” published in Front Microbiol. in 2021, authored by Megan S. Beaudry. This paper did not have any innovation or improvement in the method, but only applies to analyzing 4 equine fecal samples and 1 mock sample, with minor extension on bacteria species analysis.

2. Given the absence of innovation on methodological improvement, the paper should focus more on the analysis of the equine microbial profiling, instead of general analysis of bacteria. Although the paper begins by highlighting Gastrointestinal Tract Disorders (GITD) in horses, it fails to substantiate how the richer microbial profile uncovered by TEHC could specifically benefit the diagnosis or treatment of GITD. A more compelling approach would involve comparing fecal samples from healthy equines to those with GITD, thereby justifying TEHC's potential utility in such scenarios. Moreover, the failure of TEHC to identify Listeria, a pathogen strongly associated with intestinal illness, raises concerns regarding its clinical applicability.

3. Concerns are raised regarding the accuracy of the species-level comparison presented in Table 1.B). The data in this table requires careful reevaluation for correction.

4. Figure 3 to 5, it is better to show Figure 3, 4, and 5 as tables considering some classes in the percentage bar chart is almost invisible.

In conclusion, considering the highlighted concerns and the need for substantial revisions to strengthen the manuscript's focus, clarify its findings, and rectify inaccuracies, I recommend reconsider after major revision.

Reviewer 2 Report

Comments and Suggestions for Authors

This article describes a comparison of an older technology with a newer technology for looking at the fecal microbial makeup of the equine digestive tract. It was perhaps a bit outside my expertise in microbiome research when it comes to the methods for doing the analysis, so I do not feel like I am the best person to judge the accuracy of the statements in this paper and probably shouldn’t have agreed to review it. However, from my limited understanding of sequencing DNA, this paper did appear to be correct in the descriptions and details of the methodology. I defer to another reviewer on this matter if they have more experience than I do myself. Aside from that, I feel the paper has important information on equine microbiome research that should be published to help in the advancement toward a better understanding. The authors do a nice job of covering the methodology and of illustrating the results with abundant detail. They adequately cover the limitations of the techniques used for the analyses.

I only found minor edits in the article that should be revised. I have attached a Word document converted from the pdf so that I could incorporate my suggestions and comments. Please excuse the change in formatting that resulted from changing the file from a pdf to a Word doc, it gets a little messy but does not interfere with my edits and comments. Also, I could not find a list of the accepted abbreviations for articles, so have highlighted all the abbreviations that were not written out the first time they appear in the paper. My questions are whether GI (gastro-intestinal) and OTU (operational taxonomic units) need to be defined in the paper, and whether abbreviations can be included in the Simple Summary without explanation.
